# Leucine Zipper Downregulated in Cancer-1 Interacts with Clathrin Adaptors to Control Epidermal Growth Factor Receptor (EGFR) Internalization and Gefitinib Response in EGFR-Mutated Non-Small Cell Lung Cancer

**DOI:** 10.3390/ijms25031374

**Published:** 2024-01-23

**Authors:** Hsien-Neng Huang, Pin-Feng Hung, Yai-Ping Chen, Chia-Huei Lee

**Affiliations:** 1Department of Pathology, National Taiwan University Hospital Hsin-Chu Branch, No. 25, Ln. 442, Section 1, Jingguo Road, North Dist., Hsinchu 300195, Taiwan; ken102huang@gmail.com; 2Department and Graduate Institute of Pathology, College of Medicine, National Taiwan University, No. 1 Jen Ai Road Section 1, Taipei 100225, Taiwan; 3National Institute of Cancer Research, National Health Research Institutes, No. 35, Keyan Road, Zhunan 350401, Taiwan; hdp91111@nhri.edu.tw (P.-F.H.); 090406@nhri.edu.tw (Y.-P.C.)

**Keywords:** epidermal growth factor receptor (EGFR), non-small cell lung cancer (NSCLC), Leucine zipper downregulated in cancer-1 (LDOC1), gefitinib, clathrin adaptor protein

## Abstract

The epidermal growth factor receptor (EGFR) is a common driver of non-small cell lung cancer (NSCLC). Clathrin-mediated internalization (CMI) sustains EGFR signaling. AXL is associated with resistance to EGFR-tyrosine kinase inhibitors (TKIs) in EGFR-mutated (EGFR^M^) NSCLC. We investigated the effects of Leucine zipper downregulated in cancer-1 (LDOC1) on EGFR CMI and NSCLC treatment. Coimmunoprecipitation, double immunofluorescence staining, confocal microscopy analysis, cell surface labelling assays, and immunohistochemistry studies were conducted. We revealed that LDOC1 interacts with clathrin adaptors through binding motifs. LDOC1 depletion promotes internalization and plasma membrane recycling of EGFR in EGFR^M^ NSCLC PC9 and HCC827 cells. Membranous and cytoplasmic EGFR decreased and increased, respectively, in LDOC1 (−) NSCLC tumors. LDOC1 depletion enhanced and sustained activation of EGFR, AXL, and HER2 and enhanced activation of HER3 in PC9 and HCC827 cells. Sensitivity to first-generation EGFR-TKIs (gefitinib and erlotinib) was significantly reduced in LDOC1-depleted PC9 and HCC827 cells. Moreover, LDOC1 downregulation was significantly associated (*p* < 0.001) with poor overall survival in patients with EGFR^M^ NSCLC receiving gefitinib (*n =* 100). In conclusion, LDOC1 may regulate the efficacy of first-generation EGFR-TKIs by participating in the CMI of EGFR. Accordingly, LDOC1 may function as a prognostic biomarker for EGFR^M^ NSCLC.

## 1. Introduction

Non-small cell lung cancer (NSCLC) is responsible for 75–80% of all lung cancers and is a leading cause of cancer-related mortality [1]. The epidermal growth factor receptor (EGFR), which belongs to the ErbB family, is the most common oncogenic driver for EGFR-mutated (EGFR^M^) NSCLC. Clinical trials have demonstrated that EGFR-tyrosine kinase inhibitor (TKI) therapy is highly effective for treating EGFR^M^ NSCLC, resulting in high response rates and improved survival [2,3,4]. Mutations in the *EGFR* gene, including a small in-frame deletion in exon 19 and L858R point mutation in exon 21, accounted for 85–95% of EGFR^M^ NSCLC patients who responded to EGFR-TKIs [2,3,5]. These recurrent mutations are mapped onto the encoding region for the receptor tyrosine kinase (RTK) domain, which results in an increased sensitivity to exogenous growth factors, leading to sustained activation of EGFR signaling in tumors. These mutations are therefore referred to as EGFR-activating mutations [6]. NSCLC tumors harboring EGFR^M^ become highly dependent on the EGFR signaling pathway for growth, exhibiting an “oncogenic addiction” to the pathway [7]. First-generation EGFR-TKIs, such as gefitinib and erlotinib, reversibly bind to the ATP-binding pocket within the tyrosine kinase domain of EGFR. This blockage hinders cell proliferation and ultimately leads to cell death [8]. EGFR^M^ exhibits greater affinity for gefitinib and erlotinib than does the wild-type version [6,9], making patients harboring EGFR^M^ more responsive to EGFR-TKI therapy. Because of the higher response rates (50–70% after initial treatment) to gefitinib and erlotinib treatment among patients with EGFR^M^ than among those with EGFR^WT^, they are approved for the first-line treatment of EGFR^M^ NSCLC patients [10,11]. However, the efficacy of gefitinib or erlotinib has been limited by intrinsic and acquired resistance [12,13]. AXL is a member of the TAM RTK family and plays a key role in intrinsic and acquired resistance to gefitinib and erlotinib in EGFR^M^ NSCLC [14,15,16,17]. According to Vouri’s study, AXL can amplify EGFR signaling by interacting with EGFR in brain tumor cells [18]. The most noteworthy observation to date is that AXL is upregulated by EGFR-TKIs and induces endogenous mutators, such as components involving error-prone DNA replication, and drives the transition from drug-tolerant persister to resister in EGFR^M^ NSCLC cells [16]. Amplification of other members of the ErbB gene family, such as ErbB2 (HER2) and ErbB3 (HER3), has also been implicated in resistance to EGFR-TKIs [19]. Exogenous expression of HER2 was reported to be associated with reduced sensitivity to third-generation EGFR-TKIs in EGFR^M^ NSCLC PC9 cells [20], indicating that HER2 activity underlies the resistance of EGFR^M^ NSCLC cells to EGFR-TKIs. Additionally, observations from clinical samples revealed that it augmented HER3 in EGFR^M^ NSCLC tumors with acquired EGFR-TKI resistance [21]. According to these previous findings, we believed that identification of factors influencing activation and expression of AXL and members of the ErbB family may be useful for improving the efficacy of EGFR-TKIs and may thereby benefit patients with EGFR^M^ NSCLC.

EGFR binds ligands to its extracellular ligand-binding domains, promoting homo- and heterodimerization between EGFR and other members of the ErbB family and thereby activating their intracellular TK domains [22]. This process induces the activation of downstream signaling pathways and components involved in cell proliferation, cell cycle progression, viability, and motility [23,24]. Signal transduction of RTKs is regulated by endocytosis [25,26]. The endocytic pathway can be clathrin-dependent or independent. Clathrin-mediated endocytosis (CME) plays a crucial role in modulating EGFR signaling and is the major pathway for EGFR internalization [25]. The idea that CME plays a key role in attenuating RTK signaling is widely accepted; however, an accumulating body of evidence indicates that a blockade of internalization also decreases some RTK signaling including EGFR [27]. Sigismund’s studies demonstrated that EGFR internalized through CME is not targeted for lysosomal degradation but is instead recycled to the plasma membrane (PM) [28]. Therefore, clathrin-mediated internalization (CMI), rather than receptor degradation, is essential for sustained EGFR signaling. In CME, the clathrin adaptors, AP-1 and AP-2, connect membranes and cargo to a clathrin scaffold, controlling how specific membrane proteins, including EGFR, are internalized and recycled [29,30]. Adaptor proteins are central to CME. AP-1 and AP-2 are heterotetrameric complexes. Three of the four subunits of AP-1 and AP-2 have primary and ternary structural homologies: the large-chain β1 (AP1B1) and β2 (AP2B1), medium-chain μ1A (AP1M1) and μ2 (AP2M1), and small-chain σ1 (AP1S1) and σ2 (AP2S1). The fourth set of subunits, the large subunits γ1-adaptin of AP-1 (AP1G1) and α-adaptin of AP-2 (AP2A1), exhibit less sequence homology but have nearly identical ternary structures [29,30,31]. AP-1 mediates vesicle protein sorting between the trans-Golgi network and endosomes. Ablation of AP-1 was demonstrated to suppress EGFR recycling to the PM, regardless of the presence of epidermal growth factor (EGF) stimulation, suggesting that the function of AP-1 in recycling endosomes is to retrieve internalized EGFR to sustain cell surface expression [29]. AP-2 is the most abundant clathrin adaptor in mammalian cells; it is involved in the early phases of clathrin-coated pit nucleation and maturation and the internalization of membrane proteins [31,32]. It recruits cargo as clathrin-coated pit curvature increases and matures. This step serves as an “endocytic checkpoint” because failure to recruit cargo results in transient, abortive clathrin-coated pits [32]. During endocytosis, AP-1 and AP-2 shuttle between the PM and trans-Golgi network with clathrin [33]. Most membrane proteins undergoing CME are selected through the recognition of short linear amino acid motifs in their cytoplasmic domain by clathrin adaptors, such as the subunits of AP-1 and AP-2 [34,35]. The two major classes of peptide motifs recognized by clathrin adaptors are the tyrosine-based YXXφ (where X stands for any amino acid and φ stands for an amino acid with a bulky hydrophobic side chain) and acidic dileucine motif [E/D]xxxL[LI] [34,35]. These sequence motifs, also known as endocytic codes, function as address tags that facilitate transmembrane protein delivery and therefore effective targeting of endocytic machinery. Activated EGFR is internalized more rapidly through CME than in clathrin-independent endocytosis [28,36]. Under particular physiological conditions, such as high ligand concentration and elevated reactive oxygen species production, EGFR internalization may occur through clathrin-independent endocytosis, where the receptor is taken up by caveolae [36], or through macropinocytosis [37].

The *LDOC1* gene encodes a 146-amino-acid protein containing an N-terminal leucine zipper motif and a proline-rich region that shares a marked similarity to an SH3-binding domain (Uniprot accession number: 095751) [38]. According to the STRING database, which provides information on known and predicted protein–protein interactions, AP1M1 may interact with LDOC1 (Appendix A). Sequence analysis revealed multiple clathrin adaptor-binding motifs, including three highly conserved acidic dileucine motifs and one tyrosine-based motif, in the leucine zipper region and C-terminal domain of LDOC1, respectively (Figure 1a). We have previously demonstrated that *LDOC1* functions as a tumor suppressor gene in two EGFR-driven cancers: oral squamous cell carcinoma [39] and NSCLC [40]. LDOC1 may interact with clathrin adaptors and regulate the CMI of EGFR. In addition, LDOC1 may affect the internalization and activation of AXL, HER2, and HER3 because these molecules may bind to activated EGFR to form heterodimer complexes in NSCLC. We demonstrated that LDOC1 functions as a central regulator of EGFR internalization in EGFR^M^ NSCLC partly because it interacts with clathrin adaptors. LDOC1 depletion enhanced the activation of EGFR, HER2, HER3, and AXL, and LDOC1 downregulation was strongly correlated with poor overall survival in patients with EGFR^M^ advanced NSCLC who received gefitinib.

## 2. Results

### 2.1. LDOC1 Bound to Subunits of Clathrin Adaptor Complexes in NSCLC Cells

On the basis of the predicted interaction between LDOC1 and AP1M1 in the STRING database (Appendix A) and the presence of multiple binding motifs for clathrin adaptors in LDOC1 (Figure 1a), we speculated that LDOC1 is involved in CME of EGFR by interacting with subunits of clathrin adaptor proteins, AP1M1 and AP2M1. To enable investigation of the associations between LDOC1 and clathrin adaptor proteins, cells were serum-starved overnight and then treated with EGF (10 nM) for 30 min before being analyzed using coimmunoprecipitation (coIP)–Western blot (WB) assays. The results of this analysis support interactions of LDOC1 with AP1M1 and AP2M1 in human NSCLC cell lines, EGFR^WT^ A549, and EGFR^M^ PC9 cells (Figure 1b,c). Notably, an abundance of LDOC1-associated AP2M1 was observed in PC9 cells. To confirm these interactions, double immunofluorescence staining with confocal microscopy analysis was performed. U2OS cells were co-transfected with plasmids expressing V5-tagged LDOC1 and FLAG-tagged AP1M1 or AP2M1. After transfection, the cells were serum-starved for 6 h before EGF treatment for 30 min. Immunostaining of exogenous V5-LDOC1 (red) and FLAG-AP1M1 or AP2M1 (green) revealed widespread distribution of LDOC1 throughout the transfectants, with strong staining detected in numerous punctate structures around the nucleus and weak staining inside the nucleus (Figure 1d,e). AP1M1 and AP2M1 were predominantly localized to cytoplasmic punctate structures and colocalized with LDOC1 (Figure 1d,e; white speckles). Additionally, several adaptins were irregularly distributed on the plasma membrane and also colocalized with LDOC1 (Figure 1d,e; white dots in panels). Altogether, these results reveal associations between LDOC1 and subunits of clathrin adaptors in NSCLC cells.

### 2.2. LDOC1 Depletion Supports Either Internalization or Recycling of EGFR in EGFR^M^ NSCLC Cells

Given the importance of clathrin adaptors in CME, we propose that LDOC1 regulates CME of EGFR through competition with the receptor to bind to clathrin adaptors. To test this hypothesis, we measured the effect of LDOC1 depletion on the internalization and recycling of EGFR in two EGFR^M^ NSCLC cell lines, PC9 and HCC827, by using previously described cell surface labeling methods (Figure 2a) [29]. Cells were serum-starved overnight prior to EGF (10 nM) treatment. In the HCC827 cells, LDOC1 depletion sightly but significantly increased the internalization of phosphorylated EGFR (pEGFR), with the internalization ratio increasing from 0.9 to 1.3; however, LDOC1 depletion had no effect on the internalization of total EGFR (tEGFR). PM recycling of pEGFR and tEGFR was dramatically enhanced in the LDOC1-depleted HCC827 cells but completely abolished in the LDOC1-expressing control cells (Figure 2b). Under the same conditions, internalization of pEGFR in the LDOC1-depleted PC9 cells increased fivefold compared with internalization in the control cells, with the internalization ratio increasing from <0.1 to 0.5. Total EGFR was detected in the intracellular portion (2s) of the control cells. A trace amount of tEGFR was recycled to the PM (Re-2s) in the LDOC1-depleted PC9 cells (Figure 2c). Thus, as was the case with the HCC827 cells, LDOC1 depletion facilitated the recycling of pEGFR and tEGFR to the PM, and receptor recycling was completely inhibited in the control cells because the value of (Re-2s) was negative (Figure 2c). In total protein lysates (total), the expression of pEGFR and tEGFR in the shLDOC1 and shCtrl HCC827 cells was comparable; however, the expression of pEGFR and tEGFR in the PC9-shLDOC1 cells was higher than that in the PC9-shCtrl cells. These results, together with the aforementioned IP-WB and colocalization data (Figure 1b–e), suggest that LDOC1 depletion promotes either the CMI or recycling of EGFR through CME in EGFR^M^ NSCLC cells.

### 2.3. Depletion of LDOC1 Leads to Upregulation of the Potential EGFR-Interacting RTKs

We next explored whether the cellular transport of the potential EGFR-interacting RTKs may be regulated by LDOC1. We evaluated both total and PM levels of phosphorylated and total AXL, HER2, and HER3 in HCC827 and PC9 cell lines using the surface protein biotinylation method as described above (Figure 2a). Using this method, phosphorylated HER2 and AXL were not detected for both total and PM lysate fractions of PC9 and HCC827 cells (Figure 3). In contrast, pHER3 was primarily present in the total lysates of the two cell lines, and the levels in shLDOC1 cells were more abundant than those in control cells (Figure 3). This suggested that HCC827 and PC9 cells express cytoplasmic pHER3, independent of LDOC1 expression. Total AXL (tAXL) was observed in total and PM lysate preparations of HCC827 and total lysates of PC9 cells, and slightly increased in shLDOC1 PC9 and HCC827 cells. Similar results were obtained for total HER2 (tHER2) and HER3 (tHER3) (Figure 3). The summary of the semiquantitative data regarding total protein lysates is showed as a plot chart in Appendix A.

### 2.4. LDOC1 Downregulation Associated with Cytoplasmic EGFR Expression in NSCLC Tumors

The efficiency of the internalization and recycling processes is a key factor affecting the subcellular distribution of proteins. To examine whether LDOC1 affects the subcellular distribution of EGFR in NSCLC, we performed immunohistochemical analyses of 100 EGFR^WT^ and 100 EGFR^M^ NSCLC tumors. The clinicopathological characteristics of patients with EGFR^WT^ and EGFR^M^ advanced NSCLC stratified by LDOC1 level are presented in Table 1 and Table 2, respectively. Low LDOC1 expression was observed in 45% (45/100) and 42% (42/100) of patients with EGFR^WT^ and EGFR^M^, respectively. In EGFR^WT^ NSCLC, cytoplasmic EGFR was observed in 9.1% and 42.2% of patients with high and low LDOC1 expression, respectively. In EGFR^M^ NSCLC, cytoplasmic EGFR was observed in 12.1% and 59.5% of patients with high and low LDOC1 expression, respectively. Membranous EGFR was noticeably reduced and barely detectable in LDOC1-depleted EGFR^WT^ and EGFR^M^ NSCLC cases. Although no associations were observed between LDOC1 expression and clinicopathological characteristics, including age, sex, and cancer stage, LDOC1 depletion was significantly associated with cytoplasmic EGFR expression (*p* < 0.001) in most patients. Representative immunohistochemical images for cytoplasmic EGFR in EGFR^WT^ and EGFR^M^ NSCLC tumors are presented in Figure 4a and Figure 4b, respectively. We then investigated whether cytoplasmic EGFR was associated with NSCLC prognosis. As shown in Figure 4c,d, the survival rate for patients with tumor-expressing membranous EGFR was nonsignificantly higher than those who were expressing cytoplasmic EGFR in EGFR^WT^ and EGFR^M^ cases (*p* = 0.086 and 0.181, respectively). Collectively, the results indicated that LDOC1 downregulation was associated with cytoplasmic EGFR in EGFR^WT^ and EGFR^M^ NSCLC, possibly as a consequence of accelerated internalization.

### 2.5. LDOC1 Depletion Enhances and Sustains Prolonged Activation of EGFR, AXL, and HER2 in EGFR^M^ Cells

We demonstrated that LDOC1 may regulate the endocytosis of EGFR through CME (Figure 1 and Figure 2). Although CME has been shown to either attenuate or enhance RTK signaling, whether it modulates EGFR signaling in NSCLC remains unclear. Thus, we investigated the effect of LDOC1 depletion on the activation of EGFR and its related RTKs, including HER2, HER3, and AXL in EGFR^M^ NSCLC cells. In the PC9 cells, LDOC1 depletion approximately doubled pEGFR, pAXL, and pHER2 expression, regardless of EGF stimulation. LDOC1 depletion obviously increased the levels of pHER3 upon EGF stimulation, and tEGFR and tHER2 expression was higher in the LDOC1-depleted PC9 cells than in the control cells, regardless of EGF stimulation (Figure 5a). Similar results were obtained from HCC827 cells, but the magnitude of the increase in pEGFR, pAXL, pHER2, and pHER3 caused by LDOC1 depletion was not as dramatic as in PC9 cells under unstimulated conditions (Figure 5b). These results indicate that LDOC1 depletion enforces the activation of EGFR, HER2, HER3, and AXL in EGFR^M^ NSCLC cells. We then further examined if depletion of LDOC1 supports prolonged activation of EGFR, HER2, HER3, and AXL. To test this, we monitored changes in the expression of phosphorylated EGFR, HER2, HER3, and AXL in shCtrl and shLDOC1 PC9 cells after stimulation with EGF (10 nM) for 10 min (Figure 5c,d). The pEGFR expression in the LDOC1-expressing control cells did not further increase with EGF stimulation. However, the levels of pAXL and pHER2 in the LDOC1-expressing control cells increased, peaking at 10 min, and then sharply declined. Conversely, after EGF stimulation, expression of pEGFR in the LDOC1-depleted PC9 cells increased, peaking at 10 min, and then declined, and expression of pAXL and pHER2 increased, peaking at 10 min, but only starting to decline after 30 and 45 min, respectively. LDOC1 depletion did enhance the abundance of pHER3 but had no effect on prolonged HER3 activation. Altogether, these results demonstrate that depletion of LDOC1 not only activates EGFR, HER2, and AXL but also prolongs these RTKs’ signaling.

### 2.6. LDOC1 Downregulation Reduces Sensitivity to First-Generation EGFR-TKIs and Predicts Worse Outcomes in Patients with EGFR^M^ NSCLC Who Are Treated with Gefitinib

First-generation EGFR-TKIs, such as gefitinib and erlotinib, are the standard first-line treatments for patients with EGFR^M^ NSCLC. Given that LDOC1 depletion enhances and prolongs activation of EGFR, AXL, HER2, and HER3 (Figure 5), we believe that LDOC1 depletion affects EGFR-TKI sensitivity in EGFR^M^ NSCLC cells. We compared the effects of gefitinib, erlotinib, and osimertinib (a third-generation EGFR-TKI) on the viability of shLDOC1 and shCtrl PC9 and HCC827 cells by using MTT assays. After treatment for 72 h, the half-maximal inhibitory concentration (IC_50_) values of gefitinib and erlotinib were significantly higher in the shLDOC1 cells than in the shCtrl cells, and the IC_50_ values of osimertinib were the same in both the shLDOC1 and shCtrl PC9 cells (0.9 μM; Figure 6a) and increased from 1.8 (shCtrl) to 2.3 (shLDOC1) μM in HCC827 cells (Figure 6b). A colony-forming assay was performed to confirm the reduction in gefitinib sensitivity in the PC9 cells caused by LDOC1 depletion. Consistently, LDOC1 depletion increased the number of cell colonies in the gefitinib-treated PC9 cells (Figure 6c). Reduced sensitivity to gefitinib could lead to poorer overall survival in patients with EGFR^M^ advanced NSCLC receiving gefitinib treatment. To test this, we assessed the association between LDOC1 expression and prognosis in patients with EGFR^M^ advanced NSCLC who received gefitinib. Kaplan–Meier analysis revealed that LDOC1 downregulation was strongly associated with shorter overall survival (*p* < 0.001; Figure 6d). The results of univariate analysis indicated that the risk factors for shorter overall survival included older age (>75 years), late stage (IVA or IVB), and low LDOC1 levels (*p* = 0.003, <0.001, and <0.001, respectively), and all of these risk factors remained significant in multivariate analysis (*p* = 0.014, 0.009, and 0.002, respectively; Table 3). Notably, low LDOC1 (*p* = 0.002) was more strongly associated with shorter overall survival than late stage (*p* = 0.009). Taken together, our results show that LDOC1 depletion considerably decreased gefitinib and erlotinib sensitivity in PC9 cells. Furthermore, LDOC1 downregulation is an independent factor associated with poor overall survival in gefitinib-treated patients with EGFR^M^ advanced NSCLC.

## 3. Discussion

Although gefitinib and erlotinib is the first-line treatment for patients with NSCLC harboring EGFR^M^ in many countries, 30% to 50% of patients with EGFR^M^ do not benefit from it [10,11]. The molecular features of gefitinib-sensitive tumors must be identified, and novel therapeutic targets associated with gefitinib-resistance should be developed. This study demonstrated that LDOC1 is a critical factor affecting gefitinib efficacy (Figure 6d, Table 3). Because LDOC1 possesses multiple binding motifs for clathrin adaptors (Figure 1), it may interfere with the formation of the EGFR-AP1 and EGFR-AP2 complexes by binding to AP1M1 and AP2M1. Therefore, depletion of LDOC1 may facilitate the internalization and recycling of EGFR through CME (Figure 2b,c). In line with the findings of Sigismund [28], we found that LDOC1 depletion enhances EGFR activation (Figure 5a,b) because CMI of EGFR is not required for receptor degradation but is essential for maintaining prolonged EGFR signaling. Moreover, depletion of LDOC1 not only enhanced EGFR signaling but also augmented the activation of HER2, HER3, and AXL (Figure 5). In agreement with the importance of AXL, HER2, and HER3 to EGFR-TKI sensitivity in NSCLC [14,15,16,17,19,21], LDOC1 depletion reduced EGFR-TKI sensitivity in PC9 and HCC827 cells (Figure 6a,b). This reduced sensitivity may be why patients with LDOC1 depletion have worse prognoses than those without LDOC1 depletion (Figure 6c). Augmented activation of AXL may play a critical role in gefitinib and erlotinib resistance in LDOC1-depleted PC9 and HCC827 cells because LDOC1 downregulation causes pAXL to significantly increase, regardless of the presence of EGF (Figure 5a,b). A wealth of evidence has revealed a close relationship between AXL overexpression and EGFR-TKI resistance in NSCLC [14,15,16]. According to Noronha et al., AXL is essential for acquired resistance to EGFR-targeted therapies, such as TKIs and monoclonal antibodies [16]. Their study demonstrated that adaptive activation of AXL engages endogenous hypermutators in lung cancer cells treated with EGFR inhibitors. AXL overexpression and activation promotes survival of drug-tolerant persister cells and accelerates the emergence of T790M, an EGFR mutation specific to resistant cells. Mechanically, AXL facilitates EGFR-TKI-induced error-prone DNA polymerase and RAD18 [16]. RAD18 is an E3 ubiquitin ligase that monoubiquitinates PCNA on stalled replication forks. This enables recruitment of damage-tolerant polymerases for damage bypass and DNA repair [41]. Their study showed that AXL induced RAD18 expression in EGFR-TKI-treated NSCLC cells; hence, low-fidelity replication accelerates mutagenesis, which confers acquired resistance. Additionally, AXL induces Myc activation and unbalanced synthesis of purine, and pyrimidines also contribute to drug-resistant mutation driven by AXL. Osimertinib is a third-generation, irreversible EGFR-TKI that selectively inhibits both EGFR-TKI-sensitizing and *EGFR*-T790M-resistant mutations [42]. Notably, osimertinib has almost the same inhibitory effect on the proliferation of shCtrl and shLDOC1 PC9 cells. (Figure 6a) displays the finding that supports the assumption that mutagenesis driven by AXL may lead to the emergence of uncommon mutations of *EGFR*, such as T790M in shLDOC1 EGFR^M^ NSCLC cells, and confers resistance to gefitinib and erlotinib in LDOC1-depleted PC9 cells. In the present study, we demonstrated that LDOC1-depleted PC9 cells survived and formed colonies after long-term (10-day) treatment with gefitinib. By contrast, no colonies of PC9-shCtrl cells were observed under the same conditions (Figure 6c). These results suggest LDOC1 depletion confers acquired resistance to PC9 cells. Whether additional EGFR mutations occur in gefitinib-resistant LDOC1-depleted PC9 cells and whether AXL-induced RAD18 is responsible for these de novo mutations are topics worthy of further investigation. This study’s findings may provide a promising treatment strategy for LDOC1(−) EGFR^M^ NSCLC by using EGFR-TKIs in combination with an AXL inhibitor. In addition, because the EGFR–AXL complex has been detected in brain tumor cells [18], and EGFR can form heterodimers with HER2 and HER3, LDOC1 may also affect cellular transport, including internalization and PM recycling of these EGFR-interacting partners. In an exploratory experiment, we discovered that LDOC1 depletion increased the internalization of AXL, suggesting that interactions between EGFR and AXL may also exist in NSCLC cells.

In addition to increasing the levels of EGFR, AXL, HER2, and HER3, upregulation of cytoplasmic EGFR and reduced membranous EGFR (Figure 4b and Table 2) may contribute to reducing the efficacy of gefitinib in patients with LDOC1(−) EGFR^M^ NSCLC, given that the cytoplasmic location of EGFR can hinder drug targeting. In PC9 and HCC827 cells, LDOC1 depletion facilitates both the internalization and recycling of EGFR, and LDOC1 downregulation results in a reduction in membranous EGFR. The findings of Fraser may provide an explanation [43]. According to Fraser, removing proteins that are essential to autophagy, such as ATD7 or ATG16L1, disrupts Rab11-mediated recycling of EGFR. As a result, EGFR becomes stuck in early endosomes. Therefore, in tumor tissues, autophagy is impaired, and internalized EGFR accumulates within EGFR^M^ NSCLC cells, reducing the susceptibility of EGFR to targeted therapy.

The findings of this study have several clinical implications. Osimertinib (rather than gefitinib or erlotinib), anti-AXL monoclonal antibodies, and targeted autophagy may improve the efficacy of EGFR-targeting therapy for LDOC1(−) EGFR^M^ NSCLC. Our observations may also reveal a novel and potent strategy for overcoming LDOC1 downregulation-induced activation of members of the ErbB family and AXL. Adeno-associated viral vectors are useful in gene therapy. Clinical testing should be considered for adeno-associated viruses carrying an LDOC1 open reading frame. In summary, this study not only highlights the importance of targeting EGFR trafficking in medical oncology but also reveals a novel mechanism involving tumor suppressor genes.

## 4. Conclusions

We uncovered the endocytic codes of LDOC1, helping to elucidate how LDOC1 regulates EGFR in EGFR^M^ NSCLC. In LDOC1(+) EGFR^M^ NSCLC tumors, LDOC1 may compete with EGFR for AP-1 and AP-2 binding, preventing CMI of EGFR–EGFR complexes. Conversely, in LDOC1(−) tumors, LDOC1 depletion facilitates CMI of EGFR–EGFR complexes and possibly also EGFR–AXL, EGFR–HER2, and EGFR–HER3 dimers, which in turn sustains the prolonged activation of these members of the ErbB family and AXL. Consequently, LDOC1(−) EGFR^M^ NSCLC cells become resistant to EGFR-TKIs.

## 5. Materials and Methods

### 5.1. Patients and Tumor Biopsies

All specimens and clinical data were collected from patients who underwent a bronchoscopic biopsy, transthoracic biopsy, or surgery at National Taiwan University Hospital Hsin-Chu Branch. Formalin-fixed, paraffin-embedded tissue specimens acquired over 2012–2020 were obtained from the archives of Department of Pathology of National Taiwan University Hospital. The 200 patients included in the present study had advanced NSCLC; they comprised 100 patients with EGFR^m^ and 100 patients with EGFR^WT^ who underwent gefitinib treatment and chemotherapy, respectively. The follow-up period ranged from 0.4 to 70.5 months. Pathological sections stained with hematoxylin and eosin (HE) were reviewed and were used for the diagnosis of lung adenocarcinoma or NSCLC as World Health Organization classifications. This study was approved by the Research Ethics Committee B, National Taiwan University Hospital, Taiwan (202303103RINB). Informed consent was obtained from each patient.

### 5.2. Cell Culture, EGF Stimulation, and Reagents

A549, PC9, and U2OS cell lines were obtained from the American Type Culture Collection (Manassas, VA, USA). The HCC827 cell line was provided by Dr. Yi-Rong Chen from the Institute of Molecular and Genoomic Medicine, National Health Research Institutes, Taiwan. A549 cells were maintained in DMEM/high glucose (Cytiva, Marlborough, MA, USA). PC9 and HCC827 cells were maintained in RPMI1640 (Cytiva) [44]. The U2OS cell line was cultured in McCoy’s 5A (Modified) Medium (Gibco, Billings, MT, USA). All cells were cultured in media containing 10% FBS (Gibco), 1% penicillin (Thermo Fisher, Waltham, MA, USA), and 1% streptomycin (Thermo Fisher); the cells were incubated at 37 °C in a humidified chamber with 5% CO_2_. All cells that stably expressed siRNA-targeting LDOC1 or their scramble controls were maintained in growth medium containing 5 µg/mL puromycin (InvivoGen, San Diego, CA, USA). The cells were checked for mycoplasma (BioSmart, Seoul, Republic of Korea) upon thawing, and throughout the experimental period they consistently tested negative for mycoplasma. The cells were starved in serum-free medium overnight before EGF (10 nM) stimulation for the endocytosis assay, kinetics analysis of EGF-induced EGFR phosphorylation, and immunoprecipitation–Western blot analysis. For immunofluorescence staining, the cells were grown overnight in medium containing 0.05% FBS before EGF stimulation. Gefitinib was purchased from Selleckchem (Houston, TX, USA), and 3CAI and ASTX029 were purchased from MedChemExpress (New York, NJ, USA).

### 5.3. Plasmid Construction, Transfection, and Infection

The shLDOC1-expressing lentiviral clone, pLKO.1-shLDOC1-puro, was constructed using the LDOC1 knockdown plasmid (TRCN0000118179 containing shRNA 5′-CCGGGCTCGTGAACGAGAACCGATTCTCGAGAATCGGTTCTCGTTCAC-GAGCTTTTTG-3′) and the pLKO.1 vector provided by RNA Technology Platform and Gene Manipulation core (RNAi core), which is a research facility based in Taipei, Taiwan. The lentivirus carrying shLDOC1 pLKO.1-TRC001 was also produced by RNAi core. Through LDOC1 knockdown and the use of pLKO.1-TRC001, stable cell lines were established from PC9 and PE089 and were named PC9-shLDOC1 and PE089-shLDOC1, respectively, as per the procedure of RNAi core. PC9-shLDOC1, PE089-shLDOC1, and their corresponding control (shCtrl) cells were selected with puromycin (5 µg/mL). The Western blot analysis of whole cell lysates was performed using the customized anti-LDOC1 antibodies (Abclonal, Woburn, MA, USA), which confirmed that shCtrl cells produced LDOC1 proteins at amounts similar to those detected in parental cells, and that shLDOC1 cells produced approximately 70% less LDOC1 protein relative to shCtrl cells. Plasmid-expressing FLAG-tagged human AP1M1 and AP2M1 were purchased from OriGene (MD, USA), and plasmid-expressing V5-tagged human LDOC1 was generated using a pcDNA6.2-DEST mammalian expression vector (Thermo Fisher). All these plasmids were validated for correct ORF through DNA sequencing. U2OS cells were subcultured 16 h before transfection with lipofectamine 2000 (Invitrogen, Carlsbad, CA, USA), as per the manufacturer’s protocol.

### 5.4. Western Blot Analysis and Antibodies

Western blot analysis was performed as per the standard procedure. In brief, total protein was harvested using cell lysis buffer (Thermo Fisher) containing a protease inhibitor cocktail and phosphatase inhibitors (Thermo Fisher). The protein concentration of cellular lysates was determined using a Pierce BCA Protein Assay Kit (Thermo Fisher). Proteins were separated through SDS-PAGE and were electroblotted onto PVDF membranes (Merck Millipore, Danvers, MA, USA). The membranes were blocked for 1 h at ambient temperature with 5% skimmed milk or BSA and subsequently incubated overnight at 4 °C with primary antibodies. On the next day, the membranes were washed with TBST and incubated with HRP-conjugated goat anti-rabbit IgG (GTX213110-01) for 1 h at room temperature. The blots were developed using Clarity Western ECL Substrate (BioRad, Hercules, CA, USA). HRP-conjugated EasyBlot Anti-Rabbit IgG (GTX#221666, GeneTex, Irvine, CA, USA) was used as the secondary antibody for immunoprecipitation samples. To detect LDOC1, the mPAGE Bis-Tris SDS-PAGE Precast Gel system (Millipore) was used according to the manufacturer’s instruction. All antibodies used in Western blot analysis could be purchased from commercial companies, except for custom polyclonal anti-LDOC1 antibodies, which were produced by ABclonal. The custom polyclonal anti-pAXL^Y779^ antibodies were gifts from Dr. Chuang, S. E. (National Health Research Institutes). The antibodies used in the present study were anti-EGFR (GTX121919), anti-pEGFR^Y1068^ (GTX132810), anti-HER2 (GTX100509), HER3 (GTX100256), anti-GAPDH (GTX100118), anti-AP1M1 (GTX64907), and anti-AP2M1 (GTX#113332), which were purchased from GeneTex; anti-AXL (C89E7), which was purchased from Cell Signaling Technology; anti-pHER2^Y1248^ (AF1768-SP) and anti-pHER3^Y1262^ (AF5817-SP), which were purchased from R&D system; and anti-EGFR (51071-2-AP, Proteintech, Rosemont, IL, USA), which was purchased from ProteinTech.

### 5.5. Coimmunoprecipitation

The cells were serum-starved overnight and then stimulated with EGF (10 mM) for 30 min before protein lysates were harvested for a co-immunoprecipitation assay. Subsequently, they were washed with PBS and lysed in Pierce IP Lysis Buffer (Thermo Fisher) containing a protease inhibitor cocktail and phosphatase inhibitors (Thermo Fisher). After the cells were centrifuged at 13,000× *g* and 4 °C for 10 min, the protein concentration of the supernatant was measured, and 700–1000 μg of protein lysates were incubated with the indicated antibodies at 4 °C overnight and subsequently with protein A-magnetic beads (BioTools, Dalian, China) for 1 h at room temperature. After the immunoprecipitants were thoroughly washed, they were eluted and subjected to Western blot analysis, as per the standard procedure. The antibodies used in the co-IP experiments were custom anti-LDOC1 (ABclonal), anti-AP1M1 (12112-1-AP, Proteintech), anti-AP2M1 (144-02492, RayBiotech, Norcross, GA, USA), and anti-EGFR (51071-2-AP, Proteintech).

### 5.6. Double Immunofluorescence Staining

For the analysis of the colocalization of ectopically expressed V5-tagged LDOC1 with FLAG-tagged AP1M1 or AP2M1, U2OS cells were seeded onto coverslips and allowed to adhere overnight before DNA transfection. One day after the cells underwent transfection, they were starved overnight with medium containing 0.05% FBS. After the cells were treated with EGF (10 nM for 30 min), they were fixed with methanol at −20 °C for 10 min. Coverslips were then washed in PBS and incubated in blocking solution (5% normal goat serum in PBS) for 30 min at room temperature. The cells were then subjected to double immunofluorescence staining with the primary antibodies of rabbit pAb to V5-tag (14440-1-AP, Proteintech) and mouse mAb to FLAG-tag DYKDDDDK (66008-4-Ig, Proteintech), and the secondary antibodies of CoraLite594-conjugated goat antirabbit IgG (SA00013-4, Proteintech) and CoraLite488-conjugated goat anti-mouse IgG (SA00013-1, Proteintech). For the analysis of the association of EGFR with adaptin AP1M1 or AP2M1, A549 or PC9 cells were seeded onto coverslips and allowed to adhere overnight before they were starved overnight with medium containing 0.05% FBS. Next, fixation and blocking were performed as per the aforementioned procedures, after which double immunofluorescence staining was performed using mouse mAb to EGFR (66455-1-Ig, Proteintech) and rabbit pAb to AP1M1 (12112-1-AP, Proteintech) or AP2M1 (144-02492, RayBiotech) as primary and secondary antibodies, respectively, as per the aforementioned procedure. The immunostained cells were washed thoroughly with PBS, and their nuclei were counterstained with DAPI (100 μg/mL, Sigma-Aldrich, St. Louis, MO, USA). The stained cells were then incubated in buffer containing 0.1 M PBS, pH 8.0; 2% n-propyl gallate; and 60% glycerol, and the obtained slides were analyzed using a Leica TCS SP5 II confocal microscope (Wetzlar, Germany) and Leica LAS AF software (version 4.0).

### 5.7. Endocytosis Assay for EGFR Internalization and Recycling

Endocytosis assay was performed using a Pierce cell surface protein biotinylation and isolation kit (Thermo Fisher), as per a modified version of the method described by Uemura et al. [29]. The principle of EGFR endocytosis assay is shown in Figure 2a. In brief, the cells were serum-starved overnight. After the cells were briefly washed with ice-cold PBS, they were incubated with 0.5 mM EZ-Link Sulfo-NHS-SS-biotin (Thermo Fisher) in ice-cold PBS for 30 min. After the cells were washed twice with ice-cold PBS, they were incubated in DMEM for 30 min at 37 °C or in DMEM containing 10 nM EGF for 10 min at 37 °C to induce the internalization of biotinylated EGFR (In). Subsequently, the cells were incubated with a membrane-impermeable stripping solution (50 mM glutathione, 75 mM NaCl, 75 mM NaOH, 1% BSA, and 10 mM EDTA) on ice for 15 min to remove the remaining biotin on the surfaces of the cells (1s). The cells were re-incubated in DMEM at 37 °C for 30 min for the recycling of internalized EGFR (Re), after which they underwent a second treatment with or without the stripping solution (2s). Cellular protein lysates were harvested immediately after In, 1s, Re, and 2s. The lysates were treated with avidin-conjugated agarose for pulldown, and the recovered proteins were analyzed using Western blotting. The protein lysates were harvested immediately after biotinylation and were subjected to pulldown using avidin-conjugated agarose, then the recovered proteins were used as a PM fraction for analysis. The internalization ratio of EGFR was calculated by dividing the amount of biotinylated EGFR detected after the first stripping by that detected in the absence of the stripping reagent (1s/In). The biotinylated EGFR (pulldown) detected in step 4 (Re) was equal to the sum of the EGFR recycled to the PM and that retained within the cells (internalized and un-recycled EGFR). Thus, to obtain the ratio of recycled EGFR to internalized EGFR, the amount of biotinylated EGFR detected after the second stripping was subtracted from that detected in the absence of the stripping reagent, and the obtained value was further divided by that detected in the absence of the stripping agent (i.e., (Re−2s)/Re). The assay was verified to be performed within a range where the band intensity was linearly proportional to the protein concentration.

### 5.8. Immunohistochemistry (IHC) Staining and Hematoxylin and Eosin (HE) Staining

Formalin-fixed, paraffin-embedded tissue blocks were sliced into 5-μm sections for IHC and HE staining. In brief, slides were dipped in hematoxylin solution for 5 min; washed in distilled water; dipped in eosin solution for 2 min; and routinely processed and stained with either a customized anti-LDOC1 (ABclonal) or an anti-EGFR (EP22, Zeta, New York, NY, USA), biotinylated immunoglobulins, and a super-sensitive HRP label system as per the protocol described previously [39]. Slides with tissue sections (thickness of 5 µm) were deparaffinized with xylene and rehydrated using graded alcohols. Heat-induced antigen retrieval was performed using a citric acid buffer (pH of 6.0) for 30 min. After endogenous peroxidase activity was blocked with 0.5% H_2_O_2_ in methanol for 30 min, the sections were incubated with a customized anti-human LDOC1 antibody (1:150; Abclonal) or an anti-human EGFR (1:100; EP22, Zeta) at 4 °C overnight. After washing was completed, the specific signals on each section were developed using a BOND-PRIME Polymer DAB chromogen. LDOC1 immunoreactivity was classified as either negative (absent or weak cytoplasmic staining) or positive (moderate or strong cytoplasmic staining). EGFR immunoreactivity was evaluated and classified as membranous or cytoplasmic staining on the basis of the predominant locations of immunoreactivity. HE staining was conducted as per the manufacturer’s protocol.

### 5.9. MTT Assay for EGFR-TKI Sensitivity

Cells were seeded at a density of 5 × 10^3^ per well in a 96-well plate. The next day, the cells were treated with the indicated concentrations of gefitinib, erlotinib, and sunitinib. An appropriate amount of DMSO was also added to the control cells. After 72 h of treatment, the cells were incubated with a solution of MTT (3-(4,5-dimethylthiazol-2-yl)-2,5-diphenyltetrazolium bromide) (Sigma-Aldrich) for 4 h at 37 °C. Subsequently, the medium was aspirated, and 100 µL DMSO was added to solubilize the formazan. Colored formazan converted from MTT using viable cells was measured at 570 nm with a microplate reader (BioRad). Experiments were performed in triplicate. IC_50_ values were determined using the scientific and statistical software GraphPad Prism 10.0.0.

### 5.10. Colony Formation Assay for Gefitinib Sensitivity

The cells were seeded at a density of 3 × 10^2^ per 35-mm dish and were allowed to adhere overnight before gefitinib treatment. The cells were then incubated in medium containing varying concentrations of gefitinib at 37 °C for 10 days. Gefitinib was dissolved in DMSO, and the cells grown in medium containing 0.002% DMSO (solvent control). Colonies were counted after Giemsa staining, as per the protocol described in another study [40].

### 5.11. Statistical Analysis

In the present study, statistical analyses were conducted using the statistical software PASW Statistics (version 18.0) (IBM Corporation, Armonk, NY, USA). The clinicopathological characteristics of patients with advanced NSCLC and the associations between LDOC1 expression and various clinicopathological factors were evaluated through an X^2^ test or Fisher’s exact test. The influence of clinicopathological characteristics on overall survival was analyzed using a Cox proportional hazards model. The *p* values obtained through the Wald test were recorded. Kaplan–Meier survival analysis with a log-rank significance test was performed to estimate the probability of survival. A *p* value of less than 0.05 was regarded as statistically significant.

## Figures and Tables

**Figure 1 ijms-25-01374-f001:**
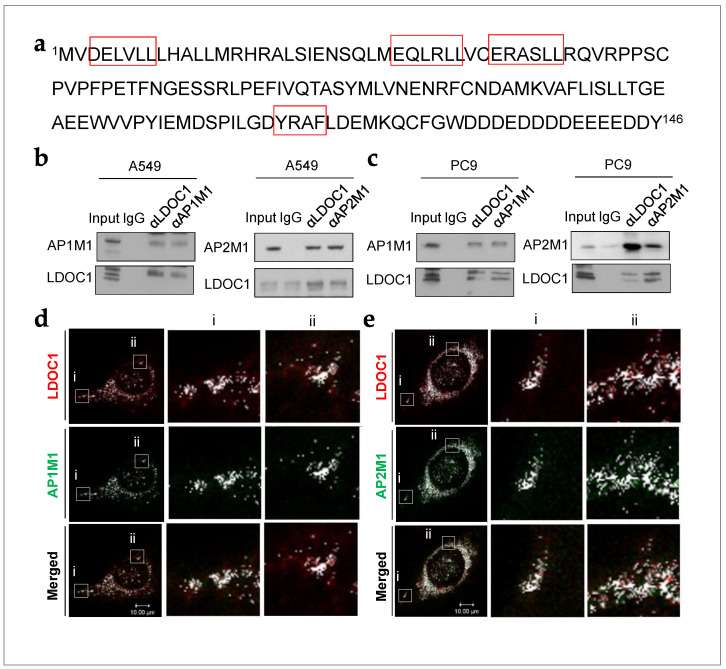
LDOC1 interacts with AP1M1 and AP2M1 in NSCLC cell lines. (**a**) Adaptin binding motifs indicated by open red box in human LDOC1. Open red box indicates binding motifs of clathrin adaptor proteins in LDOC1. (**b**,**c**) Coimmunoprecipitation and Western blot analysis revealed endogenous LDOC1–AP1M1 and LDOC1–AP2M1 interactions in A549 (**b**) and PC9 (**c**) cells. Cells were serum-starved overnight and then subjected to 10 nM EGF stimulation for 30 min. Protein lysates were then harvested and subjected to coimmunoprecipitation using antibodies against LDOC1, AP1M1, or AP2M1, followed by Western blot analysis using antibodies as indicated. (**d**,**e**) Double immunofluorescence staining analysis indicating the interactions between exogenous LDOC1, AP1M1, and AP2M1. Plasmids expressing LDOC1-V5 with AP1M1-FLAG or AP2M1-FLAG were co-transfected into U2OS cells, which were serum-starved for 6 h on the next day and then subjected to EGF (10 nM) stimulation. After EGF stimulation for 30 min, immunostaining was conducted using anti-V5 (red) and anti-FLAG (green) antibodies 24 h after transfection. Boxed regions (**a**,**b**) are magnified and presented on the right. Colocalization of LDOC1-V5, AP1M1-FLAG, LDOC1-V5, and AP2M1-FLAG presented as white speckles. Images were captured using a confocal microscope (Leica TCS SP5 II). Scale bar, 10 μm.

**Figure 2 ijms-25-01374-f002:**
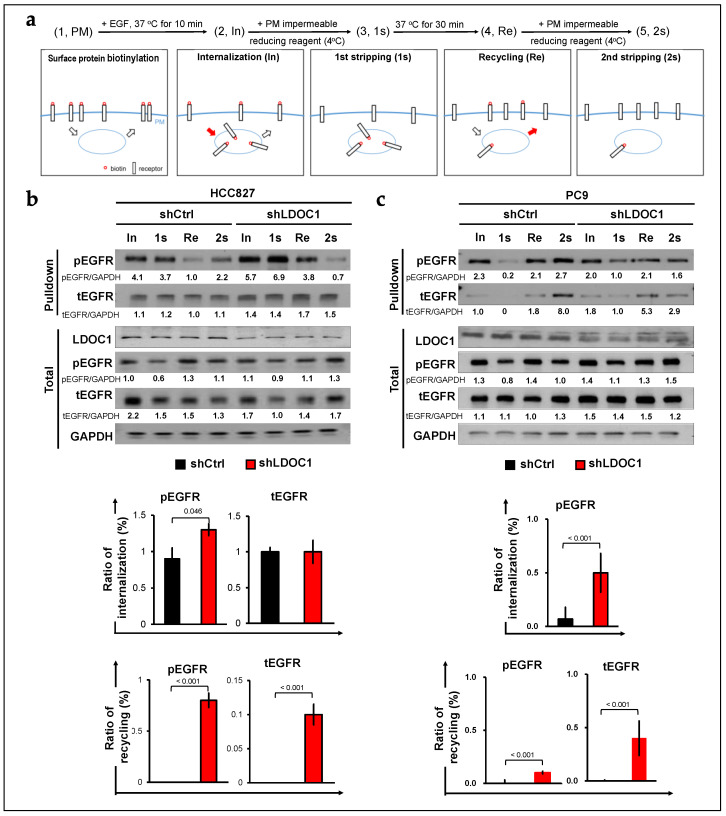
LDOC1 depletion promotes either internalization or recycling of EGFR in EGFR^M^ NSCLC cells. (**a**) Schematic of EGFR endocytosis assay completed using cell surface labelling method. Serum-starved cells were labeled with sulfo-NHS-SS-biotin and incubated with Dulbecco’s modified Eagle’s medium (DMEM) containing EGF (10 nM) for 10 min at 37 °C for internalization (In). Cells were then treated with membrane-impermeable reducing reagent to remove biotin from proteins on the plasma membrane (first stripping, 1s). Cells were re-incubated with DMEM at 37 °C for 30 min to allow for the recycling of internalized biotinylated proteins (Re) and then treated with reducing reagent (second stripping, 2s). (**b**,**c**) LDOC1-depleted (shLDOC1) and control (shCtrl) HCC827 (**b**) and PC9 (**c**) cells were treated as previously described. After internalization (In), first stripping (1s), recycling (Re), and second stripping (2s), whole-cell lysates were prepared (total), and biotinylated proteins were collected using an avidin column (pulldown). Subsequently, Western blot analysis was conducted using the indicated antibodies. Immunoblots were quantified using ImageJ (version 1.47). Ratios of internalization (1s:In) and recycling ((Re−2s):Re) were calculated and plotted as bar charts. Data are presented as mean ± SEM, *n* = 3. Differences between shCtrl and shLDOC1 cells were analyzed using Student’s *t* test.

**Figure 3 ijms-25-01374-f003:**
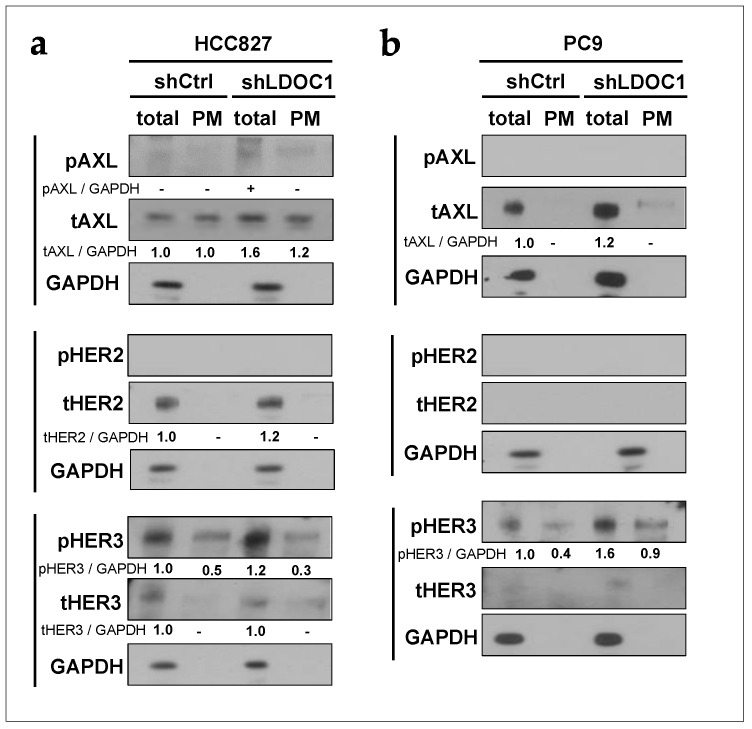
Depletion of LDOC1 causes upregulation of the potential EGFR-associated RTKs. (**a**,**b**) Western blotting of HCC827 (**a**) and PC9 (**b**) cells with (shLDOC1) and without (shCtrl) LDOC1 depletion. After biotinylation of surface proteins, cellular protein lysates (total) and the PM fraction (PM) were analyzed using antibodies against phosphorylated and total AXL, HER2, and HER3. Five percent of the input was applied in the total lane. Quantitative data of western blotting was plotted as the mean ± SD (*n* = 3) (Appendix A). Statistical differences between each shLDOC1 and shCtrl were analyzed using Student’s *t*-test.

**Figure 4 ijms-25-01374-f004:**
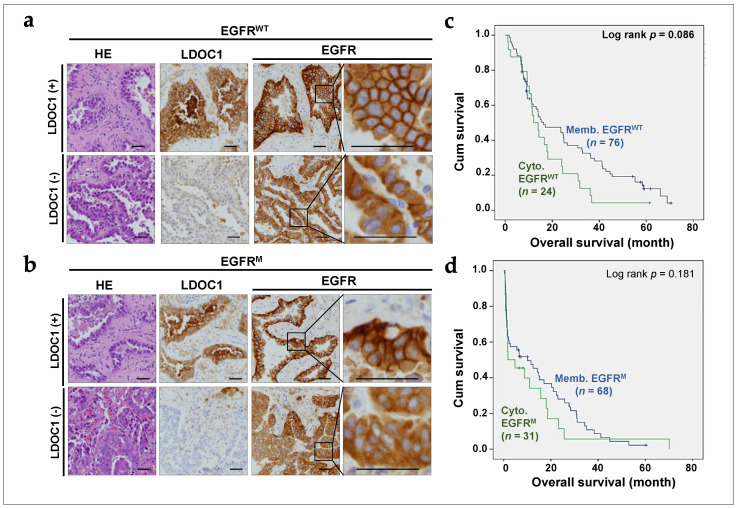
Depletion of LDOC1 reduced and enhanced membranous and cytoplasmic expression of EGFR, respectively, in NSCLC tumors. (**a**,**b**) Representative histopathological sections revealed that LDOC1 depletion was associated with diffuse cytoplasmic EGFR staining in EGFR^WT^ (**a**) and EGFR^M^ (**b**) NSCLC tumors. Scale bar = 50 μm. (**c**,**d**) Kaplan−Meier survival curves for patients with advanced EGFR^WT^ (**c**) or EGFR^M^ (**d**) NSCLC stratified into membranous (memb) and cytoplasmic (cyto) EGFR.

**Figure 5 ijms-25-01374-f005:**
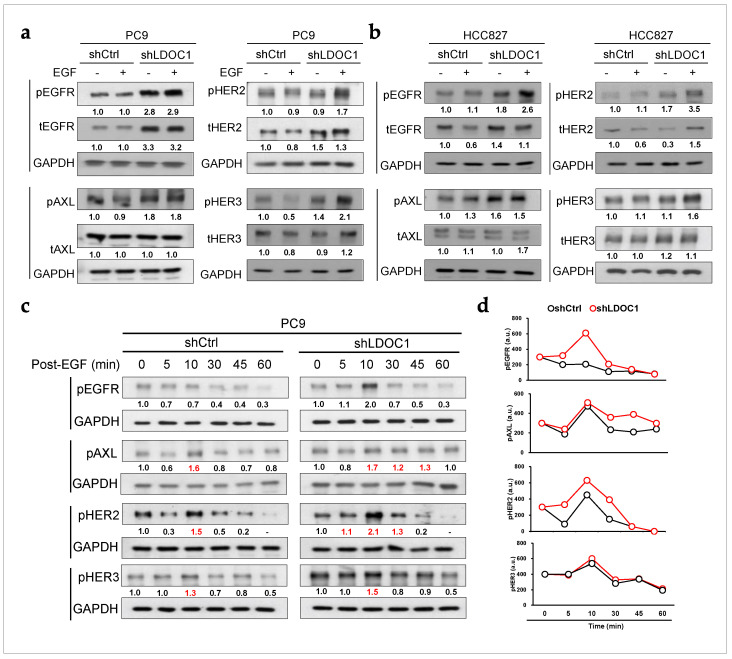
Depletion of LDOC1 promotes activation of EGFR, HER2, and AXL in EGFR^M^ NSCLC cells. (**a**,**b**) LDOC1 depletion upregulated expression of phosphorylated (p) and total (t) EGFR and its related RTKs. PC9 (**a**) and HCC827 (**b**) cells were serum-starved for 16 h before being treated with EGF (10 nM) for 30 min at 37 °C. Whole-cell lysates were subjected to Western blot analysis using the indicated antibodies. GAPDH was used as a loading control. The summary of the semiquantitative data is shown as a plot chart in Appendix A. (**c**) LDOC1 depletion sustained prolonged activation of EGFR, HER2, and AXL. PC9 cells were serum-starved overnight and then treated with EGF (10 nM) at 37 °C for 10 min, washed with PBS, and then incubated in normal growth medium at 37 °C. Whole-cell lysates were prepared at the indicated time and analyzed as described in (**a**,**b**). The value of pRTK/GAPDH at time 0 was set as 1 and fold changes > 1 were shown in red. (**d**) Semiquantitative evaluation of (**c**). Densitometry was performed on blots with different exposures and average results were plotted using arbitrary unit (a.u.) for pEGFR, pHER2, pHER3, and pAXL.

**Figure 6 ijms-25-01374-f006:**
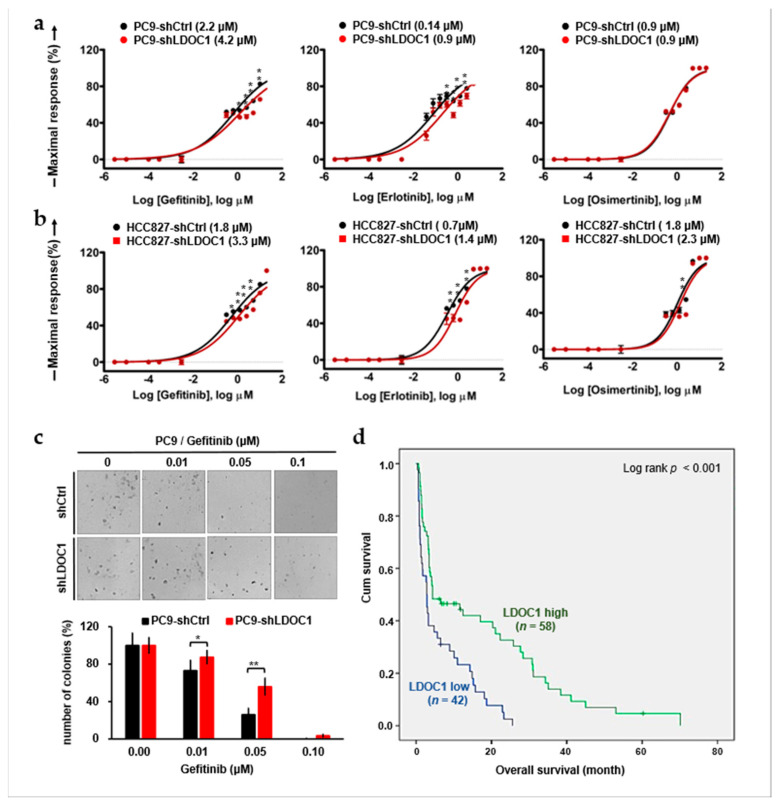
Effects of LDOC1 depletion on EGFR-TKI sensitivity and prognosis of NSCLC cells and patients, respectively, harboring EGFR^M^. (**a**–**c**) Effects of LDOC1 depletion on IC_50_ for gefitinib, erlotinib, and osimertinib in PC9 (**a**) and HCC827 (**b**) cells were determined using MTT (**a**,**b**) and colony-forming (**c**) assays. Cells were seeded in 96-well cell culture plates and treated the next day with the indicated concentrations of indicated EGFR-TKI. After 3 days, cell numbers were determined with MTT assay (**a**,**b**). Cells were seeded onto a six-well plate (300 cells per well) and treated with gefitinib. Cell colonies were counted after 10 days of incubation by using Metamorph software (version 7.7.8) and normalized using the colony numbers of the DMSO (0.02%)-treated controls. Upper: representative images of colonies grown for 10 days under indicated conditions (**c**). Data are presented as mean ± standard error of mean, *n* = 3. Differences between shLDOC1 and shCtrl at each indicated concentration were analyzed using Student’s *t* test; * *p* < 0.05 and ** *p* < 0.01. (**d**) Kaplan–Meier survival curves for patients with advanced EGFR^M^ NSCLC who received gefitinib, stratified into high (*n* = 58) and low (*n* = 42) LDOC1 expression.

**Table 1 ijms-25-01374-t001:** Clinicopathological characteristics of patients with EGFR^WT^ advanced NSCLC.

		No. (%)		
		LDOC1 High Expression	LDOC1 Low Expression	
Characteristics	No.	*n* = 55	*n* = 45	*p* Value
Age				0.933
≤75 years	36	20 (36.4)	16 (35.6)	
>75 years	64	35 (63.6)	29 (64.4)	
Gender				0.841
Male	50	27 (49.1)	23 (51.1)	
Female	50	28 (50.9)	22 (48.9)	
Stage				0.888
IIIB + IIIC	15	8 (14.5)	7 (15.6)	
IVA + IVB	85	47 (85.5)	38 (84.4)	
EGFR expression				<0.001
Membranous	76	50 (90.9)	26 (57.8)	
Cytoplasmic	24	5 (9.1)	19 (42.2)	

**Table 2 ijms-25-01374-t002:** Clinicopathological characteristics of patients with EGFR^M^ advanced NSCLC.

		No. (%)		
		LDOC1 High Expression	LDOC1 Low Expression	
Characteristics	No.	*n* = 58	*n* = 42	*p* Value
Age				0.276
≤75 years	54	34 (58.6)	20 (47.6)	
>75 years	46	24 (41.4)	22 (52.4)	
Gender				0.714
Male	45	27 (46.6)	18 (42.9)	
Female	55	31 (53.4)	24 (57.1)	
Stage				0.177
IIIB + IIIC	18	13 (22.4)	5 (11.9)	
IVA + IVB	82	45 (77.6)	37 (88.1)	
EGFR expression ^a^				<0.001
Membranous	68	51 (87.9)	17 (40.5)	
Cytoplasmic	31	7 (12.1)	24 (59.5)	

^a^ Not assessed in one case.

**Table 3 ijms-25-01374-t003:** Univariate and multivariate analyses of overall survival of patients with EGFR^M^ advanced NSCLC who received gefitinib.

		Univariate Analysis		Multivariate Analysis	
Covariate	No.	Overall Survival		Overall Survival	
		HR ^a^ (95% C.I. ^b^)	*p* Value	HR (95% C.I.)	*p* Value
Age			0.003		0.014
≤75 years	54	1		1	
>75 years	46	1.92 (1.24–2.96)		1.73 (1.12–2.67)	
Gender			0.697		
Male	45	1			
Female	55	1.09 (0.72–1.65)			
Stage			<0.001		0.009
IIIB + IIIC	18	1		1	
IVA + IVB	82	3.46 (1.63–7.35)		2.80 (1.29–6.05)	
LDOC1 expression			<0.001		0.002
High	58	1		1	
Low	42	2.36 (1.50–3.70)		2.02 (1.28–3.19)	
EGFR expression ^c^			0.181		
Membranous	68	1			
Cytoplasmic	31	1.36 (0.87–2.15)			

^a^ Hazard ratio; ^b^ confidence interval; ^c^ not assessed in one case.

## Data Availability

No new data created.

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
