# Peer review of "Leucine Zipper Downregulated in Cancer-1 Interacts with Clathrin Adaptors to Control Epidermal Growth Factor Receptor (EGFR) Internalization and Gefitinib Response in EGFR-Mutated Non-Small Cell Lung Cancer"

_ijms, 2024, doi:10.3390/ijms25031374_

Round 1
Reviewer 1 Report
Comments and Suggestions for Authors
The manuscript by Huang discusses the role of LDOC1 in clathrin mediated internalization of EGFR and potential significance of LDOC1 in EGFR signaling and patient outcomes. Based on experiments in EGFR wt A549 and EGFR mutant PC9, authors conclude that LDOC1 depletion promotes EGFR internalization and inhibits EGFR signaling by mediating EGFR–AP2M1 interaction. Furthermore, LDOC1-depleted PC9 cells become EGFR-independent. Much of conclusions in this manuscript is based on semi-quantitative analysis by western in one cell line (either wt or mutant EGFR). Therefore, it is difficult to determine the significant of findings here. Additionally, there appears to be some inconsistencies in presented data and the conclusions. For example, authors show that invasiveness is abolished by LDOC1 depletion in Figure 6d, but then Figure 7d shows patients with high LDOC1 survive better. In general, the manuscript needs major revisions. Reasoning behind many experiments is not given or are confusing. For example, it is not clear what conclusions can be drawn from Figure 3 or how a second round of internalization in Figure 2 adds to the story. Figure and legends have very low resolutions which makes them illegible to read.
Other concerns:
· Define: PM, uEGFR, pEGFR
· Not clear how EGFR connection is made here: “On the basis of the predicted interaction between LDOC1 and AP1M1 in the STRING database (Figure S1) and the presence of multiple binding motifs for clathrin adaptors in LDOC1 (Figure 1a), we speculated that LDOC1 is involved in CME of EGFR by interacting with AP1M1 and AP2M1.”
· Not clear how in Figure 2 one can conclude that “In PC9 cells, LDOC1 expression nearly abolished the recycling of pEGFR and uEGFR,”
· LDOC1 depletion was significantly associated with cytoplasmic EGFR expression (P < 0.001) in most patients. Were there some patients that were excluded?
· Data in Figure 7a is not conclusive.
· Figure 5d not labeled on the plot.
Comments on the Quality of English LanguageModerate editing of English language is required. Figures can be organized and explained better and more succinctly. Data that do not add to the story should be eliminated.
Author Response
- Some of conclusions in this manuscript is based on semi-quantitative analysis by western in one cell line (either wt or mutant EGFR)
Response 1: Thank you for your comments. There are significant differences between NSCLC cells with wild-type and mutated EFGR, particularly in sensitivity to EGFR-TKIs. To enhance experimental result consistency and bolster the conclusion, we substituted all data obtained from the EGFRWT A549 cell line with data from another EGFRM NSCLC cell line, HCC827. I hope this sufficiently answers your questions.
- Some inconsistencies in presented data and the conclusions. For example, authors show that invasiveness is abolished by LDOC1 depletion in Figure 6d, but then Figure 7d shows patients with high LDOC1 survive better
Response 2: According to genetic aberrations, NSCLC can be classified into many subtypes and each subtype exhibits distinct characteristics, such as response to EGFR TKI treatment and molecular regulator of malignancies. Therefore, it is reasonable that LDOC1 may exhibit different effects on EGFRWT and EGFRM NSCLC. The comparison of overall survival rates between high and low LDOC1 expression cohorts were make among the NSCLC patients harboring EGFRM and receiving gefitinib treatment (Fig. 7d in the previous manuscript, and is Fig. 6d in the resubmitted manuscript). Better overall survival in patients with high LDOC1 expression many reflect the higher susceptibility of control (shCtrl) cells to gefitinib as compared with LDOC1-depleted PC9 and HCC827 cells (Fig 6a,b). The transwell invasiveness assays were carried out with PC9 cells with and without LDOC1 depletion in culture medium free of gefitinib. To avoid of confusion, we have removed the invasiveness data from the resubmitted manuscript.
- it is not clear what conclusions can be drawn from Figure 3 or how a second round of internalization in Figure 2 adds to the story
Response 3: According to your comments, we have withdrawn Figure 3 and supplementary Figure S2 from the previous manuscript, and associated conclusions are also deleted.
- Figure and legends have very low resolutions which makes them illegible to read.
Response 4: I have tried to sharpen the figures. I hope you are satisfied with the quality of figures in the revised manuscript.
Other concerns:
- Define: PM, uEGFR, pEGFR
Response 5: As your request, we defined these abbreviations the first time they appear in each of three sections: the abstract; the main text; the first figure or table. Plasma membrane (PM); unphosphorylated EGFR (uEGFR); phosphorylated EGFR (pEGFR).
- Not clear how EGFR connection is made here: “On the basis of the predicted interaction between LDOC1 and AP1M1 in the STRING database (Figure S1) and the presence of multiple binding motifs for clathrin adaptors in LDOC1 (Figure 1a), we speculated that LDOC1 is involved in CME of EGFR by interacting with AP1M1 and AP2M1.”
Response 6: We have described the background and rationale for this study under the Introduction section of the manuscript. Relevant portions of your question are excerpted below. Briefly, clathrin adaptors are essential for clathrin-mediated endocytosis; hence, molecules associated with clathrin adaptors may affect the efficiency of clathrin-mediated internalization or membrane protein recycling.
“In clathrin-mediated endocytosis (CME), the clathrin adaptors, AP-1 and AP-2, connect membranes and cargo to a clathrin scaffold, controlling how specific membrane proteins, including EGFR, are internalized and recycled [29,30]. Adaptor proteins are central to CME. AP-1 and AP-2 are heterotetrameric complexes. Three of the four subunits of AP-1 and AP-2 have primary and ternary structural homologies: the large-chain β1 (AP1B1) and β2 (AP2B1), medium-chain μ1A (AP1M1) and μ2 (AP2M1), and small-chain σ1 (AP1S1) and σ2 (AP2S1). The fourth set of subunits, the large subunits γ1-adaptin of AP-1 (AP1G1) and α-adaptin of AP-2 (AP2A1), exhibit less sequence homology but have nearly identical ternary structures [29-31]. (line 84-92) …Most membrane proteins undergoing CME are selected through the recognition of short linear amino acid motifs in their cytoplasmic domain by clathrin adaptors, such as the subunits of AP-1 and AP-2 [34,35]. The two major classes of peptide motifs recognized by clathrin adaptors are the tyrosine-based YXXφ (where X stands for any amino acid and φ stands for an amino acid with a bulky hydrophobic side chain) and acidic dileucine motif [E/D]xxxL[LI] [34,35]. These sequence motifs, also known as endocytic codes, function as address tags that facilitate transmembrane protein delivery and therefore effective targeting of endocytic machinery. Activated EGFR is internalized more rapidly through CME than in clathrin-independent endocytosis (CIE) [28,36]. (line 102-111) …According to the STRING database, which provides information on known and predicted protein–protein interactions, AP1M1 may interact with LDOC1 (Fig. S1). In this study, sequence analysis revealed multiple clathrin adaptor–binding motifs, including three highly conserved acidic dileucine motifs and one tyrosine-based motif, in the leucine zipper region and C-terminal domain of LDOC1, respectively (Fig. 1a). We have previously demonstrated that LDOC1 functions as a tumor suppressor gene in two EGFR-driven cancers: oral squamous cell carcinoma [39] and NSCLC [40]. Thus, we proposed that LDOC1 may interact with clathrin adaptors and regulate CME and signaling of EGFR. (line 117-125).
- Not clear how in Figure 2 one can conclude that “In PC9 cells, LDOC1 expression nearly abolished the recycling of pEGFR and uEGFR”
Response 7: The experimental principle and mechanism of EGFR endocytosis is illustrated in Figure 2a and the Materials and Methods section. The biotinylated EGFR (pulldown) detected in step 4 (Re) equals the sum of EGFR recycled to the PM and those retained within the cells (internalized and un-recycled EGFR). Therefore, “To obtain the ratio of recycled EGFR to internalized EGFR, the amount of biotinylated EGFR detected after the second stripping (2s, EGFR remained inside) was subtracted from that detected in the absence of the stripping reagent (Re, the sum of the EGFR recycled to the PM and those internalized and un-recycled EGFR), and the obtained value was further divided by that detected in the absence of the stripping agent (Re) (i.e., [Re − 2s]/Re).” We performed the EGFR endocytosis assay with another EGFRM NSCLC HCC827 cell line to strengthen the conclusion and refined the description of the endocytosis assay results (line 196-2050 for clarity.
- LDOC1 depletion was significantly associated with cytoplasmic EGFR expression (P < 0.001) in most patients. Were there some patients that were excluded?
Response 8: Assessment of EGFR expression was failed in one patient and this patient was excluded. (line 320)
- Data in Figure 7a is not conclusive.
Response 9: We have carried out MTT assay to determine the IC50 for three EGFR-TKIs, including gefitinib, erlotinib, and osimertinib with two EGFRM NSCLC cell lines, PC9 and HCC827, to examined whether the effect of LDOC1 on the sensitivity of EGFR-targeted drugs is specific to gefitinib. I hope you are satisfied with my answer.
- Figure 5d not labeled on the plot.
Response 10: Figure 5d was not included in the previous manuscript. And the experimental results in Figure 5 have been deleted to make the manuscript more concise.

Reviewer 2 Report
Comments and Suggestions for Authors
The study investigates the role of Leucine zipper downregulated in cancer-1 (LDOC1) in non-small cell lung cancer (NSCLC) with mutated EGFR (EGFRM). LDOC1, containing binding motifs for clathrin adaptor proteins AP-1 and AP-2, was found to interact with subunits of these complexes in NSCLC cells. Depletion of LDOC1 resulted in increased internalization and proteasomal degradation of EGFR, particularly in EGFRM PC9 cells, suggesting a role in modulating EGFR-AP-2 complex formation. The study further revealed that LDOC1 depletion attenuates EGFR signaling, leading to EGFR-independent behavior and reduced sensitivity to the EGFR tyrosine kinase inhibitor gefitinib in PC9 cells. Additionally, clinical data showed a strong association between LDOC1 loss and poor overall survival in EGFRM NSCLC patients treated with gefitinib. This is an interesting study. However, points have to be addressed to enhance the interest of the readers
Major Comments
1. Why is Gefitinib used on A549 cells, given their EGFR wildtype status and predominant association with RAF/RAS mutations?
2. Authors should use additional EGFR mutant lines in addition to PC9
2. What are the effects on LDOC1 when cetuximab is used ?
3. Recent studies have revealed the impact of EGFR inhibition on compensatory mechanisms, including alterations in AXL, HER2, HER3, and other receptors. The authors should monitor the effects of LDOC1 on these receptors in their study.
4. The authors are encouraged to incorporate an additional inhibitor, such as osimertinib or erlotinib, alongside gefitinib to discern whether the observed effects are specific to the inhibitor used.
5. A recent study highlighting the AXL-EGFR regulation on RAD18 and low-fidelity polymerases has unveiled new perspectives on DNA repair (PMID: 35895872). It would be intriguing to explore whether LDOC1 similarly influences DNA repair mechanisms and the sensitivity of drugs in this context or should be discussed in the text.
Minor Comments:
1. Authors should improve the quality of all the figures. Text in figures is not readable.
2. Figure 6A and 6B. Authors can retain the Western blot images and the values of the graph can be indicated below the blot. Bar graphs can be moved to the supplementary.
3. Figure 5A and 5B, relative values have to be plotted in the line graph.
Author Response
Major Comments
- Why is Gefitinib used on A549 cells, given their EGFR wildtype status and predominant association with RAF/RAS mutations?
Response 1: Thank you for your comments. Indeed, sublines derived from the A549 cell line are not suitable for testing sensitivity to EGFR-TKIs. In the revised manuscript, all data obtained from A549 were replaced with those of an additional EGFRM NSCLC cell line, HCC827.
- Authors should use additional EGFR mutant lines in addition to PC9
Response 2: Thank you for your comments. We have carried out most of the experiments, including endocytosis assay, constitutive and EGF-induced EGFR signaling, and sensitivities to EGFR-TKIs, with an additional EGFR-mutated NSCLC cell line HCC827.
- What are the effects on LDOC1 when cetuximab is used?
Response 3: As your request, we have examined the effect of LDOC1 depletion on the sensitivity of PC9 cell line to cetuximab. The results indicated that PC9 cells are not very susceptible to cetuximab with a concentration range of 0.01~100 μg/mL, regardless of LDOC1 expression. The proliferation rates of PC9-shLDOC1 and control cells were only significantly different at 5 μg/ml cetuximab. The results are shown below:
- Recent studies have revealed the impact of EGFR inhibition on compensatory mechanisms, including alterations in AXL, HER2, HER3, and other receptors. The authors should monitor the effects of LDOC1 on these receptors in their study.
Response 4: Thank you for your constructive comment. In the revised manuscript, we monitored the effects of LDOC1 depletion on AXL, HER2, and HER3 in multiple experiments. Results indicated that either the expression or activation of AXL, HER2, and HER3 was significantly augmented by LDOC1 depletion in PC9 and HCC827 cells (Fig. 3 and 5). These findings offer additional insight into the strong association between LDOC1 expression and overall survival of patients with EGFRM NSCLC receiving gefitinib treatment. Accordingly, we modified the Introduction (line 55–67), Results (line 241–255, and line 348–374) Discussion (line 496–538) sections by adding paragraphs to describe the connections between these molecules and patients’ responses to EGFR-TKIs.
- The authors are encouraged to incorporate an additional inhibitor, such as osimertinib or erlotinib, alongside gefitinib to discern whether the observed effects are specific to the inhibitor used.
Response 5: Thank you for your insightful comment. As your request, we have examined the effect of LDOC1 depletion on the sensitivity of PC9 and HCC827 cells to additional two EGFR-TKIs, erlotinib and osimertinib, and EGFR-targeted cetuximab (Fig. 6a,b). The results suggested that resistance to EGFR-TKI due to LDOC1 depletion may be specific to first-generation EGFR-TKIs, such as gefitinib and erlotinib. Thus, the third-generation EGFR-TKI osimertinib may be an optional treatment for EGFRM NSCLC patients with low LDOC1 expression.
- A recent study highlighting the AXL-EGFR regulation on RAD18 and low-fidelity polymerases has unveiled new perspectives on DNA repair (PMID: 35895872). It would be intriguing to explore whether LDOC1 similarly influences DNA repair mechanisms and the sensitivity of drugs in this context or should be discussed in the text.
Response 6: Thank you so much for your comments. The issue of whether LDOC1 depletion led to acquired resistance to EGFR-TKIs through accelerated mutagenesis involving AXL and RAD18 is truly worthy of further investigation. Accordingly, we have modified the Introduction and Discussion sections, in line (506–530).
Minor Comments:
- Authors should improve the quality of all the figures. Text in figures is not readable.
Response 7: I have tried to sharpen the figures. I hope you are satisfied with the quality of figures in the revised manuscript.
- Figure 6A and 6B. Authors can retain the Western blot images and the values of the graph can be indicated below the blot. Bar graphs can be moved to the supplementary.
Response 8: Thank you for your suggestion. Figure 6 in the previous manuscript is now Figure 5 in the revised manuscript. I have revised the figures according to your comments. The bar graphs are presented as Supplementary Figure S3.
- Figure 5A and 5B, relative values have to be plotted in the line graph.
Response 9: To make the manuscript more concise. We have deleted the results of Fig. 5.

Round 2
Reviewer 1 Report
Comments and Suggestions for Authors
The revised manuscript by Hsien-Neng Huang is significantly easier to read and follow. This reviewer does not have any major criticism or concern. Here are a couple of minor comments:
Data in Figure 5 should be plotted in standard “dose-response” curves with zero response (100%) at zero drug concentration. Differences in drug response will be much simpler to see.
EGFR mutation status of A549 should be mentioned in the first references to these cells.
Manuscripts references “unphosphorylated” (u-) for analysis by RTK antibodies (such as uEGFR). However, these antibodies recognized total RTK (such as EGFR) and are not specific to unphosphorylated species. Therefore, these references need to be corrected.
Method section does not describe how PM lysates were isolated and collected.
Author Response
We thank you for your careful editorial review and suggestions. We have corrected the errors that you identified and respond to your comment on Figure 6 as follows.
Comment 1
Data in Figure 6 should be plotted in standard “dose-response” curves with zero response (100%) at zero drug concentration. Differences in drug response will be much simpler to see.
Answer 1
Thank you for your comments. We have revised Figure 6a and b (page 11) based on your suggestions.
Comment 2
EGFR mutation status of A549 should be mentioned in the first references to these cells.
Answer 2
We have revised the text by adding “EGFRWT” and “EGFRM” to describe the EGFR mutation status of A549 and PC9 cell lines, respectively (line 142).
Comment 3
Manuscripts references “unphosphorylated” (u-) for analysis by RTK antibodies (such as uEGFR). However, these antibodies recognized total RTK (such as EGFR) and are not specific to unphosphorylated species. Therefore, these references need to be corrected.
Answer 3
Thank you for your comments. We have corrected “unphosphorylated (u) RTKs” to “total (t) RTKs” in text and on Figure 2, 3, and 5. The corrected text is yellow highlighted in the revised manuscript.
Comment 4
Method section does not describe how PM lysates were isolated and collected.
Answer 4
We have revised the manuscript by adding the methods for isolation and collection of PM fraction in lines 233~234 and 685 ~687. Briefly, the protein lysates harvested immediately after biotinylation and were subjected to pulldown using avidin-conjugated agarose, then the recovered proteins were used as PM fraction for analysis.

Reviewer 2 Report
Comments and Suggestions for Authors
Authors have addressed the comments raised during the first round of review.
Minor Comments:
1. For cell viability dose kinetics. It will be great if the authors plot the X-Axis in a log scale for better interpretation and visualization of the data (figure 6).
2. Spell Check specifically the figures. Figure 5B is labelled as HCC927 instead of 827.
3. Figure 2b. It will be relevant to indicate that pAXL and HER2 were not detected using this method instead of absent as other methods indicated the presence of pAXL and HER2.
Author Response
We thank you for your careful editorial review and suggestions. We have corrected the typological errors and result interpretation that you identified on Figure 5b and Figure 2b, respectively. We respond to your comment and the other reviewer’s comment on Figure 6 as follows.
Comment 1
For cell viability dose kinetics. It will be great if the authors plot the X-Axis in a log scale for better interpretation and visualization of the data (figure 6).
Answer 1
Thank you for your comments. We have revised Figure 6a and b (page 11) based on your suggestions.
Comment 2
Spell Check specifically the figures. Figure 5B is labelled as HCC927 instead of 827.
Answer 2
We have corrected the labelling errors in Figure 5b (page 9).
Comment 3
Figure 2b. It will be relevant to indicate that pAXL and HER2 were not detected using this method instead of absent as other methods indicated the presence of pAXL and HER2.
Answer 3
Thank you for your comments. We have revised the manuscript on lines 234~236.
